# Timing of adverse events in patients undergoing acute and elective hip arthroplasty surgery: a multicentre cohort study using the Global Trigger Tool

Martin Magnéli [1,2] Paula Kelly-Pettersson [1,2] Cecilia Rogmark,[3,4] Max Gordon,[1,2] Olof Sköldenberg,[1,2] Maria Unbeck[1,5]

MM and PK-P contributed equally.

For numbered affiliations see end of article.

**Correspondence to**
Dr Martin Magnéli;
martin.magneli@ki.se

## ABSTRACT

**Objective** To explore timing in relation to all types of adverse events (AEs), severity and preventability for patients undergoing acute and elective hip arthroplasty.

**Design** A multicentre cohort study using retrospective record review with Global Trigger Tool methodology in combination with data from several registers.

**Setting** 24 hospitals in 4 major regions of Sweden.

**Participants** Patients ≥18 years, undergoing acute or elective total or hemiarthroplasty of the hip, were eligible for inclusion. Reviews of weighted samples of 1998 randomly selected patient records were carried out using Global Trigger Tool methodology. The patients were followed for readmissions up to 90 days postoperatively throughout the whole country.

**Results** The cohort consisted of 667 acute and 1331 elective patients. Most AEs occurred perioperatively and postoperatively (n=2093, 99.1%) and after discharge (n=1142, 54.1%). The median time from the day of surgery to the occurrence of AE was 8 days. The median days for different AE types ranged from 0 to 24.5 for acute and 0 to 71 for elective patients and peaked during different time periods. 40.2% of the AEs, both major and minor, occurred within postoperative days 0–5 and 86.9% of the AEs occurred within 30 days. Most of the AEs were deemed to be of major severity (n=1370, 65.5%) or preventable (n=1591, 76%).

**Conclusions** A wide variability was found regarding the timing of different AEs with the majority occurring within 30 days. The timing and preventability varied regarding the severity. Most of the AEs were deemed to be preventable and/or of major severity. To increase patient safety for patients undergoing hip arthroplasty surgery, a better understanding of the multifaceted nature of the timing of AEs in relation to the occurrence of differing AEs is needed.

## INTRODUCTION

Hip arthroplasty—'the operation of the century'[1] is an effective surgical treatment for both degenerative hip joint disease and femoral neck fractures. Still, adverse events (AEs) constituting a wide range of conditions

---

**STRENGTHS AND LIMITATIONS OF THIS STUDY**

⇒ This is a registry-based national multicentre study including both acute and elective hip arthroplasty patients.

⇒ Global Trigger Tool methodology for detecting adverse events was used for record review.

⇒ All hospital admissions and unplanned outpatient visits occurring up to 90 days after surgery were reviewed.

⇒ A weighted study sample optimised to select patients with adverse events was used to collect more data on adverse events compared with a random sample.

---

can occur during the primary, index hospitalisation and even years later. Both younger and older patients undergo hip arthroplasty nowadays, but the mean age of around 70 years indicates that a large group of elderly, and potentially frail patients undergo this surgery. This is particularly true for the group treated for femoral neck fractures.[2] AEs often entail suffering for the affected individual and the occurrence of preventable AEs reflects the gap between the actual care given and the expected standard of adequate safe care.[3 4]

Timing is an important factor in understanding the occurrence and prevention of AEs. It has been examined in some studies on arthroplasty surgery, but many of these included patients undergoing total primary hip or knee arthroplasty, and elective surgery was most common in these cohorts.[5–13] In contrast Bohl *et al*[10] and Malik *et al*[14] focused on geriatric patients undergoing hip arthroplasty after acute femoral neck fractures. Many of the timing studies[9–17] have used data from the American College of Surgeons National Surgical Quality Improvement

Programme with a follow-up period of 30 days postoperatively. This register does not include orthopaedic specific AEs such as dislocations.[9 12] Furthermore, many of these studies have had a rather narrow focus, for example, examining only periprosthetic joint infections,[7] pulmonary embolism,[8 18] venous thromboembolism,[19] periprosthetic femur fractures,[20] stroke,[17 21] *Clostridium difficile* colitis,[16] acute myocardial infarction[22 23] or a selection of predefined AEs.[5 9 10 12 14] In contrast, Parvizi *et al*[6] and Yao *et al*[11] have examined the occurrence of a broader range of AEs, but have not reported the timing for all the AEs that were identified.

There is a lack of knowledge regarding the timing of all types of AEs, both surgical and non-surgical, irrespective of whether they occur preoperatively, perioperatively or postoperatively in acute or elective procedures. Furthermore, there is a need to examine timing in relation to the severity and preventability of AEs. Therefore, this study aimed to explore timing in relation to all types of AEs, as well as severity and preventability in patients undergoing acute and elective hip arthroplasty.

## METHODS

### Study design
This substudy is part of a retrospective multicentre cohort study.[24] The aim of the main study was to validate the ability of a set of predefined International Classification of Diseases, 10th edition (ICD-10) codes used on a national level to compare hospitals following primary hip arthroplasties. The method and variables are the same for both the main study and this study and are briefly described below. A more detailed description has been published previously.[24 25]

### Participants and setting
The study population consisted of all patients aged ≥18 years undergoing either primary total hip arthroplasty or hemiarthroplasty, whether performed electively due to degenerative joint disease or acutely for a femoral neck fracture, that were entered into the Swedish Hip Arthroplasty Register (SHAR) during a 3-year period (N=21 774). To increase the probability of selecting primary surgery admissions with the occurrence of at least one AE and to avoid excessive record review on admissions without AEs, we used a weighted sample. The study cohort was created by combining registry data from the SHAR and the National Patient Register. In total, 20 different selection groups for acute and elective arthroplasties were created.[24] The selection groups were based on the primary length of stay and readmission which were combined with patients that had predefined ICD-10 codes indicating AEs. Larger samples were drawn from the groups that had a high risk of AE (patients with extended length of stay, readmissions and AE ICD-10 codes).

The study cohort consisted of 2000 patients who had undergone either acute or elective primary surgery. The patients had their index surgery performed in hospitals located in one of four major regions in Sweden. Two patients were excluded leaving a final study cohort of 1998 patients. The patients were followed for a maximum of 90 days postoperatively for all inpatient and acute outpatient hospital care throughout the whole country, regardless of the location of the hospital where the index surgery was performed. Therefore, the study cohort had a total of 5423 admissions in 16 of the 21 regions in Sweden, with 69 hospitals involved, as individual patients may have had multiple admissions.

### Data sources
This study used data from both medical records and three national registers, the National Patient Register, the Swedish Cause of Death Register and the SHAR. In 2019, the SHAR had a completeness of 98% and 97%, respectively, for total hip replacements and hemiarthroplasties.[2] Data on primary surgery, such as date and type of surgery, were obtained from the SHAR. These data were then cross-referenced with data from the National Patient Register using the Swedish personal identity number as a unique identifier. With these data, a timeline was created for each patient undergoing primary surgery and was used as a template to identify which admissions were to be reviewed.

We performed a manual retrospective record review using the Swedish version[26] of the Global Trigger Tool, a structured record review methodology.[27] This method involves a two-stage review process.

### Definitions
An AE was defined as suffering, physical harm or disease, as well as death related to the index admission and was not an inevitable consequence of the patient's underlying disease or treatment.

A preventable AE was defined as an event that could have been prevented if adequate actions had been taken during the patient's contact with healthcare or social care.[28] A preventable event can be related to acts of commission or acts of omission.

The index admission was defined as the orthopaedic admission during which the patient had their primary hip arthroplasty surgery.

The occurrence date of the AE was defined as either the specific date when the sign or symptom first appeared or if this information was not available and the AE occurred after discharge the date when the patient contacted the caregiver was then used. The occurrence date could also be the same date as the date of diagnosis.

### Retrospective record review process
The record review process is described in detail elsewhere.[25] The recruitment of the 10 reviewers (registered nurses, medical students and physicians) was based on previous experience of record review and/or convenience but all reviewers had previous experience of orthopaedic care.

A study manual, used as a complement to the Swedish Global Trigger Tool manual[26] with trigger definitions and

descriptions, was created to clarify the study-specific interpretations and applications of triggers, definitions and AE assessments. Triggers are clues or red flags in the medical record that indicate that a potential AE has occurred, for example, readmission within 30 days, reoperation.

The record review and time frame for the inclusion of an AE covered the period from the start of the admission to a maximum of 90 days postoperatively. During the follow-up period, it was possible to identify AEs in the records of patients receiving inpatient and/or outpatient care, irrespective of the specialty, at any hospital in Sweden.

The reviews were carried out in a two-stage process using standardised data collection forms, one form for each review stage. In most cases, the same person carried out both stages of the review and the data were entered into a study-specific database.

In the first review stage, all record entries from all healthcare professionals were reviewed. The reviewers screened for the presence of one or more of the 38 predefined triggers in five modules of the Swedish version of the Global Trigger Tool.[26] Only records with triggers indicating at least one potential AE went forward to review stage 2.

In review stage 2, the reviewers sorted the different triggers into potential AEs, as more than one of the triggers can be involved in a single AE. Every potential AE was then reviewed separately. To qualify as an AE, a score of 3 or higher on a 4-point Likert scale was required (1=the AE was not related to index admission, 2=the AE was probably not related to index admission, 3=the AE was probably related to index admission and 4=the AE was related to index admission). The preventability was also assessed using a similar scale with the same cut-off limit for inclusion regarding preventability.

Several other variables were collected, for example, the timing, type and severity of the AEs. Severity was determined using a slightly modified version of the National Coordinating Council for Medication Error Reporting and Prevention (NCC MERP) Index.[29] NCC MERP Index categories E–I were included as they are related to harm, that is, AE. Events determined to be risks and no-harm incidents, that is, NCC MERP categories A–D, were excluded.

### Patient and public involvement

This study did not involve any patient or public representatives.

### Statistical methods

Time from surgery to AE occurrence is presented in median days and IQR. The statistical analysis was performed using R V.4.1.0 (R Project for Statistical Computing, Vienna, Austria), using tidyverse (V.1.3.1) for dataset manipulation, ggplot2 (V.3.3.5) for plots and htmlTable (V.2.2.1) for creating tables.

## RESULTS

### Demographics

One-third of the cohort were patients admitted acutely with a femoral neck fracture and two-thirds underwent surgery on an elective basis. Patients with acute admissions were older, received hemiarthroplasties more often and were more often treated at university hospitals (table 1). Elective hemiarthroplasty represents a subgroup of patients, who all had the indication 'status post femoral neck fracture (FNF)' or avascular necrosis.

**Table 1** Demographics

| | All patients n=1998 | Acute admitted n=667 | Elective admitted n=1331 |
|---|---|---|---|
| Sex | | | |
| Female, n (%) | 1250 (62.6) | 444 (66.6) | 806 (60.6) |
| Male, n (%) | 748 (37.4) | 223 (33.4) | 525 (39.4) |
| Age, median years (min–max, IQR) | 77 (18–100, 16) | 84 (34–100, 10) | 73 (18–99, 16) |
| LOS, median days (min–max, IQR) | 6 (1–56, 5) | 7 (1–56, 8) | 5 (1–52, 3) |
| Type of surgery | | | |
| Total arthroplasty | 1435 (71.8) | 143 (21.4) | 1292 (97.1) |
| Hemiarthroplasty | 563 (28.2) | 524 (78.6) | 39 (2.9) |
| Type of hospital, n (%) | | | |
| University | 630 (31.5) | 295 (44.2) | 335 (25.2) |
| Central County Council | 556 (27.8) | 180 (27.0) | 376 (28.2) |
| County Council | 531 (26.6) | 109 (16.3) | 422 (31.7) |
| Private* | 281 (14.1) | 83 (12.4) | 198 (14.9) |

Weighted samples, the values are not representative for average Swedish orthopaedic care concerning hip arthroplasty.
*Most of the surgeries are publicly financed through agreements with the regional authorities.
LOS, length of stay.

## AE outcomes

In total, 2116 AEs of varying severity were identified in 1171 (58.6%) patients in our weighted sample of which 527 (45.0%) of these patients were affected by more than one AE. The patients admitted acutely sustained 981 (46.4%; min to max, 1–10) AEs and the elective patients 1135 (53.6%; min to max 1–7). Acute patients were affected by an AE to a higher extent compared with elective patients, 71.4% (476/667) vs 52.3% (696/1331). The patients in the cohort had in total 3425 all-cause readmissions during the follow-up period. However, some of the readmissions were not related to the index admission, and were therefore, not considered to be related to an AE.

## Timing overall and in connection to AE types

Four AEs did not have a correct date registered and were excluded, leaving 2112 AEs remaining for the timing analysis, 980 AEs in acute patients and 1132 AEs in elective patients. Nineteen (0.9%) AEs occurred from admission to the day before surgery and 2093 (99.1%) AEs occurred from the day of surgery, including perioperative AEs, up to 90 days postoperatively.

Of the 2112 AEs, 970 (45.9%) occurred during the index orthopaedic admission, the remaining 1142 (54.1%) AEs occurred after discharge. The AEs occurring after discharged were distributed as follow: 866 (41.0%) within 30 days and 276 (13.1%) between 31 and 90 days after surgery. Of the patients undergoing acute surgery 486 (49.6%) of the 980 AEs occurred during the index admission, compared with elective patients where 484 (42.8%) of 1132 AEs occurred during the index admission.

The median time from day of surgery to an AE was 8 days for the 2093 AEs occurring on the day of surgery and postoperatively. The corresponding median for both acute and elective patients was also 8 days (summary in table 2, full table in online supplemental material).

For both acute and elective patients, pressure ulcers, skin, tissue and superficial vessel harm, perioperative and postoperative bleeding/haematomas not requiring reoperation and pneumonia were common AEs and peaked during the index admission. Within 30 days after surgery, but after discharge, dislocation of the prothesis and infections such as deep periprosthetic infections, superficial wound infections and urinary tract infections were commonly occurring AEs and peaked in this period for both groups. No AE peaked after 30 days postoperatively for the acute patients, whereas for the elective patients, pulmonary embolisms, deep venous thromboses, mechanical complications and surgical harm—other all peaked after 30 days. Some types of AEs continued to occur at a high rate such as dislocations in both groups after 30 days (table 2).

The eight most common types of AEs and their median day of occurrence are shown in figure 1. Of these, perioperative/postoperative bleeding/haematoma had the earliest day of occurrence at 3 (IQR, 12) days for acute patients and 0 (7) days for elective patients. The second earliest occurring AE was skin, tissue and superficial vessel harm with a median of 5 (9) for acute and 4 (4) days for the elective patients. Dislocations had the longest median time to occurrence with 23 (24 vs 29) days for both groups.

## Timing outcomes within 5 days postoperatively

In total, 40.2% of both minor and major AEs (n=842/2093) occurred either on the day of surgery or during the following 5 postoperative days (table 3). Some types of AEs were only common early in the 5-day time frame, for example, perioperative haemorrhage, perioperative fractures and dislocation on postoperative day 1 while some other AEs such as skin, tissue and superficial vessel harm and distended urinary bladder were frequent from the day of surgery and onwards. Later in the time frame, infections such as pneumonia, urinary tract infections and superficial wound infections began to occur more commonly.

## Timing and preventability in relation to severity

Preventability and timing varied within and between the acute and elective groups, as well as, for the different AE types regarding severity. Most of the AEs were deemed to be of major severity (n=1370, 65.5%, NCC MERP categories F–I) or preventable (n=1591, 76.0%). The proportion of major AEs was higher among the elective patients compared with the acute, 72.9% vs 56.8%. In contrast, the proportion of AEs that contributed to death was higher among the acute patients, 3.9% vs 0.5%. All deaths in the elective group were deemed preventable and the AEs contributing to death began to occur later (median days 20.5) in contrast to the 65.9% preventable AEs for the acute group with a median of 15 days (table 4). Median days to death was in total 29 days for 28 acute patients and 25 days for 4 elective patients. Examples of AEs resulting in some degree of permanent harm, classified according to the Swedish patient insurance assessment standards, were dislocations, deep periprosthetic infections, fractures, thromboses/embolisms and leg length difference.

In total, one-third of the AEs were classified as minor (NCC MERP category E) including, for example, pressure ulcers, urinary tract infections, skin harm, falls with minor injury, distended urinary bladders and intravenous infiltrations. In total, minor AEs and AEs that required intervention necessary to sustain life within 60 min (NCC MERP category H) occurred earlier (median 4 respective 1 day) compared with the other NCC MERP categories. Among the acute patients, minor AEs occurred more often, although somewhat later and were assessed to be more preventable compared those in elective patients (table 4).

## DISCUSSION

To the best of our knowledge, this is the first study to explore all types of AEs with associated timing data, both in total and separately for acute and elective patients.

**Table 2** Summary of identified adverse events for acute respective elective patients, sorted by descending order of types of AEs for all patients

| Type of AEs | All | | Acute | | Elective | |
|---|---|---|---|---|---|---|
| | N (%) | Median time from surgery to AE (days) | n (%) | Median time from surgery to AE (days) | n (%) | Median time from surgery to AE (days) |
| Dislocation of the prothesis | 274 (13.1) | 23 | 101 (10.4) | 23 | 173 (15.4) | 23 |
| Pressure ulcer | 189 (9) | 5 | 143 (14.8) | 5 | 46 (4.1) | 4.5 |
| Urinary tract infection | 163 (7.8) | 8 | 93 (9.6) | 9 | 70 (6.2) | 8 |
| Periprosthetic joint infection | 149 (7.1) | 20 | 36 (3.7) | 19 | 113 (10) | 20 |
| Superficial wound infection | 147 (7) | 11 | 59 (6.1) | 11 | 88 (7.8) | 13 |
| Skin tissue and superficial vessel harm* | 123 (5.9) | 4 | 78 (8.1) | 5 | 45 (4) | 4 |
| Perioperative/postoperative bleeding/haematoma—did not require reoperation | 117 (5.6) | 0 | 29 (3) | 3 | 88 (7.8) | 0 |
| Pneumonia | 117 (5.6) | 4 | 77 (8) | 3 | 40 (3.6) | 5 |
| Falls | 87 (4.2) | 15 | 38 (3.9) | 19 | 49 (4.4) | 10 |
| Neurological† | 87 (4.2) | 2 | 46 (4.8) | 2.5 | 41 (3.6) | 2 |
| Distended urinary bladder | 81 (3.9) | 3 | 52 (5.4) | 3 | 29 (2.6) | 3 |
| Pulmonary embolism | 64 (3.1) | 27 | 15 (1.5) | 12 | 49 (4.4) | 32 |
| Gastric ulcer | 45 (2.2) | 6 | 19 (2) | 7 | 26 (2.3) | 5.5 |
| Cardiovascular‡ | 40 (1.9) | 3 | 14 (1.4) | 13.5 | 26 (2.3) | 3 |
| Pain | 36 (1.7) | 4 | 7 (0.7) | 3 | 29 (2.6) | 9 |
| Gastrointestinal§ | 31 (1.5) | 7 | 9 (0.9) | 7 | 22 (2) | 6.5 |
| Perioperative fracture | 31 (1.5) | 0 | 8 (0.8) | 0 | 23 (2) | 0 |
| Renal failure | 31 (1.5) | 2 | 20 (2.1) | 2 | 11 (1) | 3 |
| Deep vein thrombosis | 29 (1.4) | 42 | 10 (1) | 24.5 | 19 (1.7) | 46 |
| Allergic reaction | 25 (1.2) | 5 | 8 (0.8) | 5 | 17 (1.5) | 4 |
| Leg length difference | 23 (1.1) | 3 | 4 (0.4) | 2.5 | 19 (1.7) | 3 |
| Gastrointestinal infection | 22 (1.1) | 9 | 17 (1.8) | 11 | 5 (0.4) | 4 |
| Septicaemia | 20 (1) | 17.5 | 14 (1.4) | 17.5 | 6 (0.5) | 13.5 |
| Mechanical complication¶ | 20 (1) | 25 | 7 (0.7) | 24 | 13 (1.2) | 29 |
| Unclear infection | 17 (0.8) | 5 | 11 (1.1) | 5 | 6 (0.5) | 4 |
| Respiratory | 17 (0.8) | 1 | 9 (0.9) | 1 | 8 (0.7) | 1 |
| Myocardial infarction | 15 (0.7) | 7 | 6 (0.6) | 3.5 | 9 (0.8) | 8 |
| Mouth and throat infection | 12 (0.6) | 9 | 8 (0.8) | 9 | 4 (0.4) | 10 |
| Peripheral nerve injury** | 12 (0.6) | 2 | 3 (0.3) | 8 | 9 (0.8) | 2 |
| Ileus | 9 (0.4) | 5 | 4 (0.4) | 6 | 5 (0.4) | 4 |
| Stroke | 9 (0.4) | 4 | 4 (0.4) | 5 | 5 (0.4) | 4 |
| Electrolyte imbalance | 8 (0.4) | 5.5 | 1 (0.1) | 5 | 7 (0.6) | 6 |
| AE caused by anaesthesia†† | 7 (0.3) | 1 | 3 (0.3) | 0 | 4 (0.4) | 1 |
| Malnutrition | 7 (0.3) | 6 | 4 (0.4) | 13 | 3 (0.3) | 6 |
| Bleeding–not related to surgery‡‡ | 5 (0.2) | 11 | 2 (0.2) | 6.5 | 3 (0.3) | 11 |
| Infection other | 5 (0.2) | 11 | 4 (0.4) | 11 | 1 (0.1) | 6 |
| Bleeding—that required reoperation | 5 (0.2) | 8 | 1 (0.1) | 8 | 4 (0.4) | 5 |
| Surgical harm—other | 3 (0.1) | 71 | 0 (0) | NA | 3 (0.3) | 71 |
| Multiorgan failure | 1 (0) | 3 | 1 (0.1) | 3 | 0 (0) | NA |

Continued

**Table 2** Continued

| Type of AEs | All | | Acute | | Elective | |
|---|---|---|---|---|---|---|
| | N (%) | Median time from surgery to AE (days) | n (%) | Median time from surgery to AE (days) | n (%) | Median time from surgery to AE (days) |
| Other | 10 (0.5) | 7.5 | 3 (0.3) | 19 | 7 (0.6) | 7 |
| Total | 2093 (100) | 8 | 968 (100) | 8 | 1125 (100) | 8 |

Examples of AEs in these types; full table available in online supplemental material.
*Blister, extravasation, phlebitis.
†Acute confusion, hallucination, lethargy.
‡Heart failure.
§Obstipation, vomiting, diarrhoea.
¶Fracture without a fall, reoperation after several dislocations.
**Foot drop.
††Awareness, aspiration.
‡‡In connection to urinary catheter or warfarin treatment.
AE, adverse event.

Most of the AEs occurred after the index admission in both groups. As much as 87% of the AEs occurred within 30 days postoperatively. The overall median time of 8 days from the day of surgery to an AE was the same for both acute and elective patients. Pressure ulcers and skin, tissue and superficial vessel harm were common and peaked for both group during the index admission. Dislocations and deep periprosthetic joint infections were common reasons for readmissions within 30 days after surgery and peaked during this time frame. Dislocations continued to occur to a high extent after 30 days. Forty per cent of all AEs, both major and minor, occurred from the day of surgery and up to and including the fifth postoperative day. The timing and preventability varied regarding the

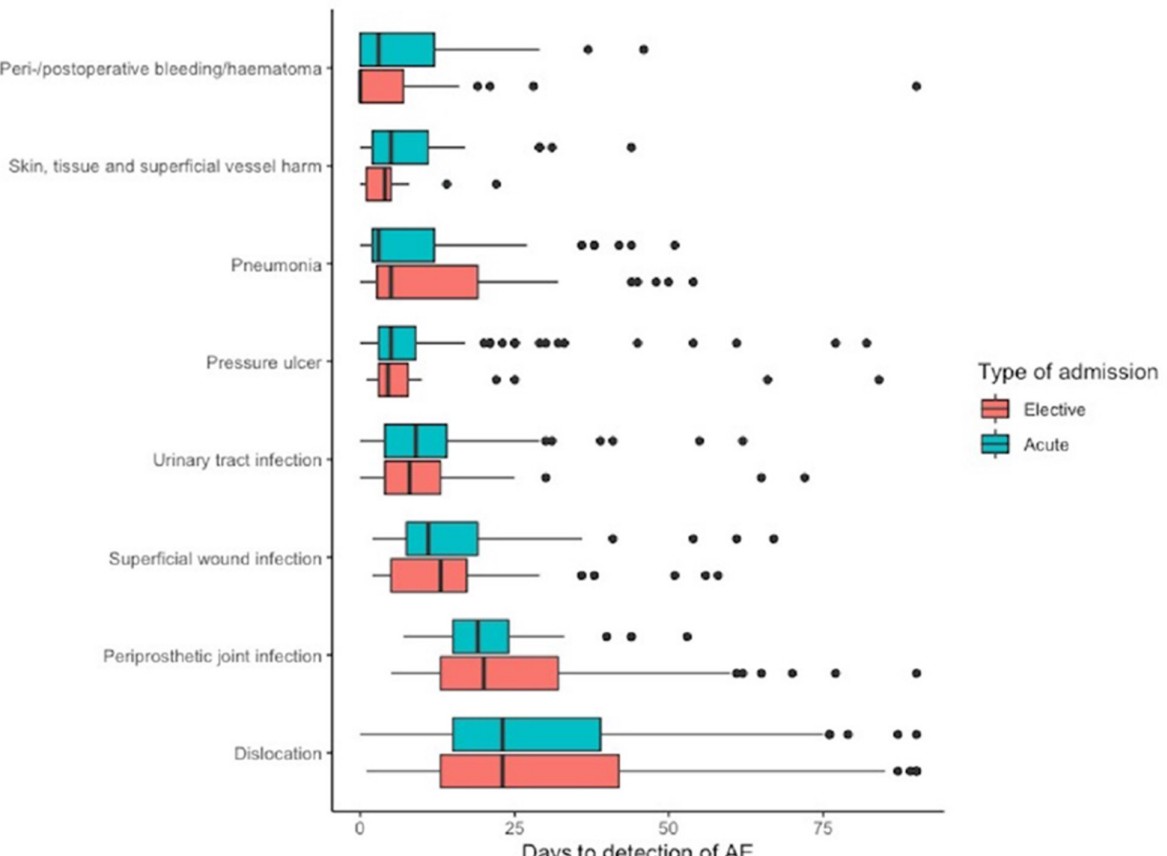

**Figure 1** Median days from the day of surgery for the eight most common types of adverse events (AEs) up to 90 days postoperatively.

**Table 3** Perioperative and early postoperative adverse events (AEs), the five most common AE types per day

| Type of AE, n | POD 0 | POD 1 | POD 2 | POD 3 | POD 4 | POD 5 |
|---|---|---|---|---|---|---|
| Perioperative haemorrhage | 59 | | | | | |
| Perioperative fracture | 25 | | | | | |
| Skin, tissue and superficial vessel harm | 11 | 14 | | 17 | 10 | 11 |
| Neurological | 9 | 20 | 18 | | | |
| Distended urinary bladder | 8 | 9 | 16 | 17 | 8 | |
| Dislocation | | 14 | | | | |
| Pressure ulcer | | 14 | 21 | 30 | 17 | 15 |
| Pneumonia | | | 27 | 18 | | 7 |
| Urinary tract infection | | | 14 | 11 | 13 | 5 |
| Superficial wound infection | | | | | 12 | 5 |
| Total for the five most common per day, (%) of total per day | 112 (64.0) | 71 (48.6) | 96 (56.8) | 93 (58.5) | 60 (55.0) | 43 (51.2) |
| Total for all AEs per day | 175 | 146 | 169 | 159 | 109 | 84 |

POD, postoperative day.

severity. Most of the AEs were deemed to be preventable and/or of major severity.

Previous timing studies examining patients undergoing arthroplasty have used varying data collection methods, included varying types of arthroplasty surgery, often with a focus on total hip replacements in elective patients, and the criteria for AEs, timing and follow-up periods differ. Furthermore, some studies do not report data for hip arthroplasty separately. This makes comparison of the outcomes somewhat difficult.

## Type of AEs and timing in general

A wide variability was found regarding the timing of AEs. Perioperative/postoperative bleeding/haematoma, perioperative fracture, AEs caused by anaesthesia, respiratory AEs, renal failure, peripheral nerve injuries and neurological AEs occurred early, within the first 2 days postoperatively. The latest occurring AEs included surgical harms of other types, deep vein thromboses, mechanical complications, dislocations and periprosthetic infections

**Table 4** Number, proportion and timing of adverse events (AEs) and preventable AEs in relation to severity

| | Severity according to NCC MERP | | | | | |
|---|---|---|---|---|---|---|
| Groups | E | F | G | H | I | Total |
| AEs, all patients | | | | | | |
| N (%) of AEs | 723 (34.5) | 794 (37.9) | 518 (24.7) | 14 (0.7) | 44 (2.1) | 2093 (100) |
| Median (IQR) days | 4 (7) | 8 (15) | 21 (27) | 1 (3) | 15.5 (18.75) | 8 (16) |
| N (%) of preventable AEs | 528 (73.0) | 541 (68.1) | 482 (93.0) | 11 (78.6) | 29 (65.9) | 1591 (76.0) |
| Median (IQR), days, preventable AEs | 4 (7) | 8 (14) | 21 (27) | 1 (2.5) | 16 (18) | 9 (17) |
| AEs, acute patients | | | | | | |
| N (%) of AEs | 418 (43.2) | 335 (34.6) | 169 (17.5) | 8 (0.8) | 38 (3.9) | 968 (100) |
| Median (IQR) days | 5 (8) | 7 (13) | 22 (23) | 1 (1.75) | 15 (20.25) | 8 (14) |
| N (%) of preventable AEs | 321 (76.8) | 243 (72.5) | 155 (91.7) | 8 (100) | 23 (60.5) | 750 (77.5) |
| Median (IQR), days, preventable AEs | 5 (7) | 7 (12.5) | 22 (20.5) | 1 (1.75) | 15 (19.5) | 8 (14.75) |
| AEs, elective patients | | | | | | |
| N (%) of AEs | 305 (27.1) | 459 (40.8) | 349 (31.0) | 6 (0.5) | 6 (0.5) | 1125 (100) |
| Median (IQR) days | 3 (6) | 8 (16) | 21 (31) | 1.5 (6.25) | 20.5 (13) | 8 (18) |
| N (%) of preventable AEs | 207 (67.9) | 298 (64.9) | 327 (93.7) | 3 (50.0) | 6 (100) | 841 (74.8) |
| Median (IQR), days, preventable AEs | 3 (5) | 8.5 (15.75) | 21 (29) | 1 (14.5) | 20.5 (13) | 10 (19) |

E, contributed to or resulted in temporary harm; F, contributed to or resulted in temporary harm that prolonged hospitalisation, required outpatient care or readmission; G, contributed to or resulted in permanent patient harm; H, required intervention necessary to sustain life within 60 min; I, contributed or resulted in the patient's death.
NCC MERP, National Coordinating Council for Medication Error Reporting and Prevention Index.

which ranged from median 20 to 71 days. Some AEs continued to occur up until the end of the follow-up period.

Dislocation was the most common AE in elective patients and the second most common in acute patients. It occurred at a median of 3 weeks and in 9 of 10 cases after discharge. This AE is seldom reported in timing studies, for example, the commonly used National Surgical Quality Improvement Programme does not include dislocations and other orthopaedic-specific AEs.[9 12] However, Ali et al[30] also found that dislocation was one of the most common AEs which clustered early after discharge. In their study, after 30 days postoperatively 74% of the dislocations requiring readmission had occurred and this was higher than the 51% in this study. Even after 30 days, dislocations continued to occur at a high rate.

Other common AEs were infections such as periprosthetic, urinary tract and superficial wound infections, the majority of which occurred after discharge. The median time to superficial wound infections and periprosthetic infections was 11 vs 20 days and the corresponding for Malik et al[14] was 16 vs 23 days. Bohl et al[9 10] reported in two studies surgical site infections with median days of 17 and 16, respectively. Our median time to urinary tract infections is in line with results from other studies with a range of 7–8 days.[9 10 14]

In contrast to other AEs, pulmonary embolism and deep vein thrombosis occurred relatively late at 4 and 6 weeks, respectively. The longer time to the occurrence of deep vein thrombosis compared with pulmonary embolism can also be seen in other studies.[5 9 10 14] In contrast, these two AE types occurred later in our cohort than in other studies where median days to pulmonary embolism and deep vein thrombosis ranged from 3 to 5 days and 6 to 9 days, respectively.[5 9 10 14] A partial explanation can be that we had a longer follow-up period, which affects the median day outcomes.

Several of our most common AE types, such as pressure ulcers, skin, tissue and superficial vessel harm, falls, neurological harm and distended urinary bladder, are not covered by similar timing studies examining AEs in hip arthroplasty patients. These AEs, which mostly peaked during the index admission, were often classified as minor and preventable but not directly related to the surgical intervention. Nevertheless, minor AEs may also lead to suffering for the patient. To reduce the number of AEs occurring in orthopaedic care, it is important to analyse AEs also from a timing perspective and work proactively preoperatively, perioperatively and postoperatively in interprofessional teams both inside and outside the operating room.[31]

Two timing studies focused on geriatric patients with hip fractures in general[14] and those treated with arthroplasty.[10] Although, timing of specific AEs was quite similar, pulmonary embolism, deep vein thrombosis and sepsis occurred somewhat later and superficial wound infections occurred somewhat earlier in our acute cohort. The patterns of AE types in relation to early or late onset were similar for the most part.

## Timing during and after index admission, early and late onset

We found that less than half of all AEs occurred during the index admission. The median length of index stay is longer in our study compared with several other studies. This can be explained by the fact that we included acute patients and admissions with longer length of stay were over-represented due to the sampling technique used in the main study. Nevertheless, other studies with shorter length of stay found that most of the AEs occurred during the index admission.[9 10 12] In contrast, Yao et al[11] had a higher AE occurrence after discharge when studying a cohort of home-admitted patients. As we are moving toward shorter hospital stays and an increased amount of day surgery, the risk of major AEs occurring outside the hospitals increases. Therefore, a careful follow-up plan after discharge can be of value. Coproduction of healthcare outcomes by the healthcare professionals and the patient as an important participating partner may be a significant intervention in the preventive work to reduce the occurrence of AEs and increase patient safety.[32] The patients, and in some cases their significant others and the staff at nursing homes, need to be well informed regarding which signs and symptoms of AEs to be aware of and how to manage self-care. These are central aspects during the rehabilitation period.

Most of the AEs in this study occurred within 30 days postoperatively. Many other studies had a follow-up period of only 30 days and most of the AEs would probably be detected during this period. However, this time frame may lead to a risk of underestimation of the incidence of AEs and a bias towards the earlier-diagnosed AEs as some AE types, for example, surgical site infections did not plateau within this specific follow-up period.[9 10 14]

Nearly half of all AEs occurred within the first 5 days postoperatively. Still, none of the four 'catastrophic' AEs, predefined by Johnson et al[12] namely pulmonary embolism, myocardial infarction or cardiac arrest, cerebrovascular accident or death were among our five most common AEs during this time period. The corresponding results for Parvizi et al[6] was 93% for life-threating medical AEs. Belmont et al's[13] findings showed that 77% of total hip arthroplasty patients who had an adverse cardiac event in the form of myocardial infarction or cardiac arrest experienced it within 3 days after surgery.

## Severity and preventability

The timing and preventability varied regarding the severity for the acute and elective patients. We assessed severity with a widely used scale in AE studies using the Global Trigger Tool record review methodology.[27] The severity of each AE was assessed independently, and a specific type of AE, for example, pressure ulcers, falls and urinary tract infections could be assessed as having varying degrees of

severity depending on the harm caused. Parvizi *et al*[6] and Yao *et al*[11] classified all, for example, pneumonia as minor AEs while we classified them as major. On the other hand, Parvizi *et al*[6] defined AEs that resulted in, for example, prolonged hospital stays as minor while we classified these AEs as major (NCC MERP F).

In contrast to other studies,[5 6 11–14] we did not include death as a separate AE type as we considered death to be the consequence of an AE and belongs to the severity classification. The median time to death was around 4 weeks for both groups. For some patients more than one AE may have contributed to their death (NCC MERP category I).

Further research is needed to explore the multifaceted nature of the timing in relation to the occurrence of minor and major AEs, both surgical and non-surgical, during the preoperative, perioperative or postoperative periods of patient care. A multipronged approach is required in the prevention of AEs,[14] as most AEs were deemed to be preventable, and patients were commonly affected by more than one AE. To increase patient safety for this patient group, additional research is needed regarding the implementation and evaluation of timely targeted interventions during the varying phases of the patient-care process, focusing on preventable AEs.

## Limitations and strengths

The study has several limitations. We have not included AEs occurring after 90 days postoperatively. Using record review as the data collection method may have led to an underestimation of AEs due to under-reporting. Furthermore, AEs detected and treated outside the hospital setting may have been missed. These are probably minor AEs as severe AEs would be more likely to be treated in a hospital setting. This has led to severe outcome, for example, periprosthetic joint infections being more common compared with superficial wound infections. Even though we reported the time to occurrence in most cases, some AEs occurring after discharge may have had a delayed timing date if the patient was not specific enough about when the first symptom of the AE occurred. In these cases, the date when the patient first contacted the caregiver was used. The weighted study sample used in this study was useful for maximising the inclusion of patients with an AE in relation to the main study's aim and this is the explanation for the high AE rates. This is not the incidence of AEs in the source population. The estimated incidence for different AEs has been calculated by adjusting for the sampling weights in previous publications from this study.[24]

The multicentre design with a wide range of patients of all ages and types of hospitals can be considered as a strength. The 90-day national follow-up, regardless of index hospital should be sufficient to detect most acute, subacute and rare AEs. To ensure the reliability and validity of our data, the study has been closely monitored to check the correctness and completeness of the data, enabling for a good control of the review process. We used a stringent definition of what constitutes an AE and attained very good kappa values in the review process.[24] We only included AEs related to the care given and excluded conditions judged to be related to an underlying disease or condition, unless the disease or condition deteriorated due to healthcare. We believe our results to be valid and generalisable at least to Western world patients and/or publicly financed healthcare.

## Author affiliations
[1]Department of Clinical Sciences at Danderyd Hospital, Karolinska Institute, Stockholm, Sweden
[2]Department of Orthopaedics, Danderyd University Hospital, Danderyd, Sweden
[3]Department of Clinical Sciences Malmö, Clinical and Molecular Osteoporosis Research Unit, Lund University, Lund, Sweden
[4]Department of Orthopaedics, Skåne University Hospital Malmö Orthopedics Clinic, Malmo, Sweden
[5]Högskolan Dalarna, Falun, Sweden

**Acknowledgements** The authors wish to thank Marie Ax, Susanne Hansson, Zara Hedlund, Mirta Stupin, Tim Hansson, Lovisa Hult-Ericson and Christina Jansson for their help and support with the review records. We would also like to thank all departmental managers for providing access to the patient records and Per Nydert for his help with the study database.

**Contributors** MM, guarantor, collected data, analysed the data and critically revised the manuscript. PK-P interpreted the data and contributed to the drafting of the work. CR contributed to the design of the study and contributed to the drafting of the work. MG contributed to the design of the study, collected data and critically revised the manuscript. OS contributed to the design of the study, collected data and critically revised the manuscript. MU contributed to the design of the study, monitored the reviews, analysed and interpreted the data, and was mainly responsible for the drafting of the work. All authors have approved the final version of the manuscript and agree to be accountable for all aspects of the work.

**Funding** This work was supported by the Swedish Patient Insurance.

**Disclaimer** The funder has not been involved in any part of the study, in either writing the manuscript or the decision to submit the manuscript for publication.

**Competing interests** None declared.

**Patient and public involvement** Patients and/or the public were not involved in the design, or conduct, or reporting, or dissemination plans of this research.

**Patient consent for publication** Not applicable.

**Ethics approval** This study involves human participants and was approved by Regional Ethics Committee of Gothenburg ID DNR: 516-13 and T732-13. Registry-based study without the need to collect informed consent.

**Provenance and peer review** Not commissioned; externally peer reviewed.

**Data availability statement** No data are available. Data sharing is prohibited by local regulations.

**ORCID iDs**
Martin Magnéli http://orcid.org/0000-0003-0341-0227
Paula Kelly-Pettersson http://orcid.org/0000-0003-3685-2953

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
