## [Reviewer comments · BMJ Open]

ARTICLE DETAILS

TITLE (PROVISIONAL)	Timing of adverse events in patients undergoing acute and elective hip arthroplasty surgery: a multi-centre cohort study using the Global Trigger Tool
AUTHORS	Magnéli, Martin; Kelly-Pettersson, Paula; Rogmark, Cecilia; Gordon, Max; Sköldenberg, Olof; Unbeck, Maria

VERSION 1 – REVIEW

REVIEWER	Magill, Paul Musgrave Park Hospital
REVIEW RETURNED	30-Jul-2022

GENERAL COMMENTS	Congratulations on a huge body of work. I agree it is an important and neglected topic. I think a few things would benefit from clarification. 39 Elective admissions for hemiarthroplasty? 13% (173/1331) dislocation rate in elective group? 8% (113/1331) deep infection rate in elective group? Deep infection more common than superficial? These stats give me concern that your population is not representative of normal or else there is a problem with data collection. Can you reassure the reader? If you have the data I would also be interested to see how many patients represented to the ED and how many patients required re admission for any cause within 30 days. Both of these stats are brought up when day case arthroplasty is discussed. We have a 2% rate of readmission for any cause within 6 weeks following day case THR (not published). I don't know if this is acceptable as I'm not aware of a reliable comparison to non-day case arthroplasty.
--

REVIEWER	Keihanian, Faeze Mashhad University of Medical Sciences
REVIEW RETURNED	15-Nov-2022

GENERAL COMMENTS	This study is well-written by Magnéli et al. and evaluated the timing of adverse events in patients undergoing acute and elective hip arthroplasty surgery: a multi-center cohort study. There are just some typos and grammar errors that should be corrected. Using the below citations is suggested: doi: 10.2147/ORR.S215240. eCollection 2019. doi: 10.2147/ORR.S184590. eCollection 2019.
---

VERSION 1 – AUTHOR RESPONSE

Reviewer: 1

Dr. Paul Magill, Musgrave Park Hospital

Comments to the Author:

Congratulations on a huge body of work.

I agree it is an important and neglected topic.

I think a few things would benefit from clarification.

Answer: Thank you for these kind and encouraging words!

Question: 39 Elective admissions for hemiarthroplasty?

Answer: Thank you for this question. We can understand that this seems strange, and therefore we have performed an analysis of the participants with elective hemiarthroplasties. This is a subgroup of patients, who all had the indication "Status post femoral neck fracture (FNF)" or avascular necrosis. Individuals with failed internal fixation will often receive a hemiarthroplasty. Non-displaced FNF (Garden 1 & 2) are usually fixed with pins or screws in Sweden. 80% of the study participants with hip fractures received a hemiarthroplasty, and 20% a total hip arthroplasty. The mean age was 84 for those with elective hemiarthroplasty, indicating that they would have been treated with a hemiarthroplasty if they had sustained a displaced FNF. None of them had osteoarthritis as the indication for surgery.

Action:

We added "Elective hemiarthroplasty represent a subgroup of patients, who all had the indication "Status post femoral neck fracture (FNF)" or avascular necrosis." To comment this on page 8, line 11 & 12.

Question: 13% (173/1331) dislocation rate in elective group?

8% (113/1331) deep infection rate in elective group?

These stats give me concern that your population is not representative of normal or else there is a problem with data collection. Can you reassure the reader?

Answer: We understand your concerns. You are correct that the study sample is not a representative sample. This high rate of adverse events (AEs) is due to the weighted sample design of the study. We selected patients with readmissions and ICD-codes indicating AEs to have a more efficient study and avoid excess record review on medical records without AEs. The sampling process is described in detail in previous publications. The incidence in the source population can be estimated by adjusting by the weights in the sampling groups. The estimated incidence for different AEs has been calculated by adjusting for the sampling weights in previous publications from this study. However, the focus of this paper is when AEs occurs, and we have therefore not included any adjusted incidence rates.

Action: We realise that this was not clear enough in the manuscript and we have now revised the manuscript accordingly by adding more information (page 5, line 14-19) and adding "The weighted study sample used in this study was useful for maximising the inclusion of patients with an AE in relation to the main study's aim and this is the explanation for the high AE rates. This is not the incidence of AEs in the source population. The estimated incidence for different AEs has been

calculated by adjusting for the sampling weights in previous publications from this study” on page 17, line 24-28.

Question: Deep infection more common than superficial?

Answer: This is also an interesting remark, thank you for highlighting this. The explanation is again the study design. We assume that patients with superficial wound infections will normally be treated as outpatients, whilst those with peri-prosthetic joint infections will almost always be admitted to hospital for surgical intervention. The record review was limited to hospital admissions and Emergency Department visits. Therefore, the study will detect most peri-prosthetic joint infection cases. As most superficial wound infections will be treated by planned outpatient visits or General Practitioner visits, they will therefore not be detected to the same degree as peri-prosthetic joint infections.

Action: We have made a clarification about this on page 17 “Furthermore, AEs detected and treated outside the hospital setting may have been missed. These are probably minor AEs as severe AEs would be more likely to be treated in a hospital setting. This has led to severe outcome, for example, peri-prosthetic joint infections being more common compared to superficial wound infections.”

Question: If you have the data I would also be interested to see how many patients represented to the ED and how many patients required re admission for any cause within 30 days. Both of these stats are brought up when day case arthroplasty is discussed. We have a 2% rate of readmission for any cause within 6 weeks following day case THR (not published). I don't know if this is acceptable as I'm not aware of a reliable comparison to non-day case arthroplasty.

Answer: We do have this data. However, the rates are not representative and comparable with a random sample. The weighted sample used in this study will favour the selection of patients with adverse events. Day case surgery was rare in Sweden during the study period and only 6 participants had a length of stay (LOS) of 1 day in our cohort. Still, we believe that the LOS for our cohort was very short compared to the preceding decades and this was the idea behind the prediction model we developed on the cohort data. Please see Magnéli et al. Measuring adverse events following hip arthroplasty surgery using administrative data without relying on ICD-codes. <https://doi.org/10.1371/journal.pone.0242008>. Here we found that a slight increase in LOS is predictive of an AE because LOS is pressed to the minimum. Readmissions were also predictive of AEs.

Action: none.

Reviewer: 2

Dr. Faeze Keihanian, Mashhad University of Medical Sciences

Comments to the Author:

This study is well-written by Magnéli et al. and evaluated the timing of adverse events in patients undergoing acute and elective hip arthroplasty surgery: a multi-center cohort study.

Question: There are just some typos and grammar errors that should be corrected. Using the below citations is suggested:

doi: 10.2147/ORR.S215240. eCollection 2019.

doi: 10.2147/ORR.S184590. eCollection 2019.

doi:10.2147/TCRM.S155918. eCollection 2018.

doi: 10.1097/MD.0000000000008235.

Answer and action: Thank you for reading our manuscript and your kind words. We have revised the manuscript for typos and grammatical errors. Also, thank you for the suggested studies, which address many interesting aspects of arthroplasty surgery. We could not, regrettably, make any place for them in the current manuscript.

VERSION 2 – REVIEW

REVIEWER	Magill, Paul Musgrave Park Hospital
REVIEW RETURNED	13-Mar-2023

GENERAL COMMENTS	I now appreciate that this is a paper based on trigger tool methodology. I think it would be better to include that in the title. The current title has the potential to mislead the reader who may, like me, extrapolate the results to the whole population. I am now also beyond my level of comfort and am not familiar with using the trigger tool methodology. While to me it appears to be a well run study, I suggest an epidemiologist read it.
--

VERSION 2 – AUTHOR RESPONSE

Dear editor,

On the behalf of all the authors, I would like to thank you and the reviewers for the opportunity to revise this manuscript again. We agree with reviewer 1, that the title might be strengthened and the aim might be clearer by adding information about the Global Trigger Tool. We have revised the title to: "Timing of adverse events in patients undergoing acute and elective hip arthroplasty surgery: a multi-centre cohort study using the Global Trigger Tool". We have attached both an unmarked and a marked copy (changed text in red). We hope that you will consider publishing this paper after this revision.

Sincerely
Martin Magnéli, MD, PhD